# Hybridogenesis in the Water Frogs from Western Russian Territory: Intrapopulation Variation in Genome Elimination

**DOI:** 10.3390/genes12020244

**Published:** 2021-02-08

**Authors:** Ikuo Miura, Vladimir Vershinin, Svetlana Vershinina, Andrei Lebedinskii, Alexander Trofimov, Ivan Sitnikov, Michihiko Ito

**Affiliations:** 1Amphibian Research Center, Hiroshima University, Higashi-Hiroshima 739-8526, Japan; 2Institute of Applied Ecology, University of Canberra, Canberra 2601, Australia; 3Institute of Natural Sciences and Mathematics, Ural Federal University, Ekaterinburg 620026, Russia; 4Institute of Plant and Animal Ecology, Russian Academy of Sciences, Ural Division, 620144 Ekaterinburg, Russia; s_verchok@list.ru (S.V.); alexandertrofimov92@gmail.com (A.T.); ivan11011994@gmail.com (I.S.); 5Institute of Biology and Biomedicine, Lobachevsky State University, 603950 Nizhny Novgorod, Russia; leb-nn@yandex.ru; 6School of Science, Kitasato University, Sagamihara 252-0373, Japan; michichikoito.ito@gmail.com

**Keywords:** *Pelopylax kl. esculentus*, genome introgression, cytochrome b, *Serum albumin*

## Abstract

Hybridogenesis in an interspecific hybrid frog is a coupling mechanism in the gametogenic cell line that eliminates the genome of one parental species with endoduplication of the remaining genome of the other parental species. It has been intensively investigated in the edible frog *Pelophylax kl. esculentus* (RL)*,* a natural hybrid between the marsh frog *P. ridibundus* (RR) and the pool frog *P. lessonae* (LL). However, the genetic mechanisms involved remain unclear. Here, we investigated the water frogs in the western Russian territory. In three of the four populations, we genetically identified 16 RL frogs living sympatrically with the parental LL species, or with both parental species. In addition, two populations contained genome introgression with another species, *P. bedriagae* (BB) (a close relative of RR). In the gonads of 13 RL frogs, the L genome was eliminated, producing gametes of R (or R combined with the B genome). In sharp contrast, one RL male eliminated the L or R genome, producing both R and L sperm. We detected a variation in genome elimination within a population. Based on the genetic backgrounds of RL frogs, we hypothesize that the introgression of the B genome resulted in the change in choosing a genome to be eliminated.

## 1. Introduction

“Biological species” is the concept of species proposed by Ernst Mayr [1]. He defined species as “*groups of actually or potentially interbreeding natural populations, which are reproductively isolated from other such groups*”. The reproductive isolation mechanisms between species include gamete isolation, hybrid lethality and hybrid sterility in post-mating. Hybrid lethality is the death of hybrids during early development. Hybrid sterility indicates normal development based on the body plan except the gonads; meiosis or gametogenesis does not completely proceed and fails to produce fertile gametes. Such reproductive isolating mechanisms are considered to be caused by genomic incompatibility between the two species in the somatic cells and/or germ cells. However, the nature of “genomic incompatibility” remains unclear [2].

Hybridogenesis in an interspecific hybrid mainly occurs in fish and frogs [3,4,5,6,7,8]. The hybrids develop normally and produce fertile gametes comprising either genome of the parental species. One genome set of the parental species is eliminated in the germ cells of the gonads. This system implies that the genomes of both species are precisely recognized with each other to allow either genome to be eliminated from a single cell. Therefore, a specific signature is engaged in the genome of each individual species. Thus, hybridogenesis is an excellent subject for improving our understanding of genome differentiation in establishing reproductive isolation during the evolution of one species to another.

The edible frog, *Pelophylax kl. esculentus* (RL, a species genotype), engages a hybridogenetic system. This species is a natural hybrid between the marsh frog *P. ridibundus* (RR) and the pool frog *P. lessonae* (LL). It produces fertile gametes through either the system of *lessonae-esculentus* (L-E) or *ridibundus-esculentus* (R-E) [6,7,9,10]. In the L-E system, where the *P. kl. esculentus* lives with LL species, the L genome is eliminated and the remaining R genome is endoduplicated to form R gametes [11,12,13,14]. On the other hand, in the R-E system, where *P. kl. esculentus* lives with RR species, the R or L genome is eliminated and either or both gametes are produced from one hybrid [12,15,16,17]. Are the two systems stable or changeable with each other? What genetic factor is involved in choosing the genome to be eliminated?

In this study, to approach the above questions, we investigated the water frogs from western Russian territory, of which populations are located 800 km west from the eastern end of their geographic distribution (Figure 1). To date, few genetic studies have included the populations [18] and the unusual situation of hybridogenesis in the water frog complex is expected to be brought about by immigration of levant frog *P. bedriagae* into the Russian territories [19,20,21]. Our study identified a variation in the genome elimination within one population.

## 2. Materials and Methods

### 2.1. Collecting Frogs and Taxonomic Identification

A total of 108 adult frogs were collected from five locations in the middle of the East European plane of Russia in May to September during 2017 to 2019 (Figure 1). The five locations are classified into four different populations (P1–P4) based on the taxonomic composition: the two locations in P3 population are distant from each other but are combined to one population because the two are located within a city and both do not include any *P. kl. esculentus*. The location, sex, and species are listed in Appendix A. Species was identified molecularly based on the genotype of *Serum albumin* intron 1 (SAI-1); *P. ridibundus SA* includes a transposable element within the intron 1, while *P. lessonae* does not (Figure 1) [22]. The SAI-1 genotype was determined by polyacrylamide electrophoresis (PAGE) of the amplified fragments. In addition, identification of *P. kl. esculentus* was confirmed based on the heterozygous sequences of the *Sox3* and *Rhodopsin* genes (Appendix A). The sex of the specimens was determined based on the external morphology, such as black thumb callus of males and eggs of females, or by inspection of gonads after euthanasia (soaked in 0.1–1.0% MS222). Animal care and experimental procedures were conducted with the approval of the Committee for Ethics in Animal Experimentation at Hiroshima University (Permit Number: G17-3-3).

### 2.2. Artificial Crossing and Microscopic Observation of Gonads

Ovulation was induced in females by injection of LHRH (Salmon, 4013835, Bachem AG, Bubendorf BL, Switzerland) dissolved in Holtfreter’s solution according to the method of Berger et al. [23], and the eggs were artificially inseminated with sperm from the male according to the method of Ohtani et al. [24]. For microscopic observation of gonads (with Nikon Eclipse 80i/Plan Fluor, DS-US/DS-Ri1 camera), gonads were cut out and fixed with Nawashin fixing solution (27% formalin). Paraffin sections (10 μm) were double-stained with hematoxylin and eosin.

### 2.3. Cytochrome b and Serum albumin Gene Analyses

Total genomic DNA was extracted from finger clip tissues or gonads of the frogs using NucleoSpin Tissue (TaKaRa, Kusatsu, Japan), according to the manufacturer’s instruction. To amplify the fragment of mitochondrial *cytochrome b (Cyt b)* and nuclear *serum albumin* intron 1 genes, PCR was performed using EmeraldAmp PCR master mix (TaKaRa, Kusatsu, Japan) as follows: 1 μL of DNA solution was amplified in 25 μL reaction volume containing 12.5 μL 2-fold Premix including Taq polymerase and dNTP and 0.5 μL of each of the 12.5 μM primers at 94 °C for 1 min for one cycle, 55 °C for 20 s and 72 °C for 30 s or 1.2 min for 35 cycles using Smart cycler (TaKaRa, Kusatsu, Japan). The primers we designed for this study were: Cyt b-f 5′CTCCTRGGAGTCTGCC3′ and Cyt b-r 5′agrtctttgtaggara3′ for *cytochrome b*, and SAI-1-f 5′TGTACTGGCGACCCTA3′ and SAI-1-r 5′ctgcctttacaatatc3′ for *Serum albumin* intron 1. The amplified fragments were purified using FastGene^TM^ Gel/PCR Extraction Kit (Nippon Genetics, Tokyo, Japan) and the nucleotide sequence was determined using an ABI PRISM 3130xl genetic analyzer (Applied Biosystems) according to the manufacturer’s instructions. The alignments of nucleotide sequence and construction of gene tree were performed using MEGAX software [25]. Sequence data from this article have been deposited with the DDBJ data libraries under Accession Nos. LC599369~LC599378 for *cytochrome b* and LC599379~LC599385 for *Serum albumin* intron 1. The sequence data of *cytochrome b* (MG214959.1, KY613559.1, KU158348.1 and AJ880677.1) and *Serum albumin* intron 1 (FN432363.1, FN432364.1, FN432365.1, FN432372.1, LN794317.1 and FN432377.1) were obtained from GenBank.

For genotyping of SAI-1, the amplified product was electrophoresed in 6% acrylamide gel with 1× TBE (Tris, Boric Acid and 2Na-EDTA) buffer with Quick-Load purple 1 kb plus DNA ladder (New England Biolabs, Ipswich, MA, USA), and was photographed using UV illuminator after staining with ethidium bromide.

## 3. Results

### 3.1. Taxonomic Composition

To elucidate the taxonomic composition in each of the Russian populations investigated (Figure 1), we molecularly identified the three species (*Pelophylax kl. esculentus*, *P. lessonae* and *P. ridibundus*) based on the genotype of the *Serum albumin* intron 1 (SAI-1) (see Materials and Methods and Figure 1). The results are provided in Figure 2 and Appendix A. In all four populations, *P. lessonae* was predominant in the constitution (56.2–87%, while *P. ridibundus*, 0–43.8%). *P. kl. esculentus* was identified in the three populations of P1, P2 and P4 (11.7–25.7%). P3 only contained *P. lessonae* and *P. ridibundus*. In P2, all three species were combined.

### 3.2. Mitochondrial Origins

To elucidate the cytoplasmic origins of *P. kl. esculentus*, we determined the mitochondrial *cytochrome b (Cyt b)* sequence. The *Cyt b* of all *P. kl. esculentus* from P4 and one individual from P2 (P2–3) constituted one clade with *P. lessonae* in all four populations. The haplotype of *P. lessonae* from Sweden was included (Figure 3). On the other hand, *Cyt b* of two and three *P. kl. esculentus* individuals from P1 and P2, respectively, constituted another clade with all *P. ridibundus* from P2 and six *P. ridibundus* individuals from P3, including the haplotype of *P. ridibundus* from Poland. The remaining clade comprised one *P. kl. esculentus* from P1 and *P. ridibundus* from P3 and included the haplotype of *P. bedriagae* (a close relative of *P. ridibundus*) (Figure 3). We conclude that these Russian populations of *P. kl. esculentus* possess three different cytoplasmic origins: *P. lessonae*, *P. ridibundus* and *P. bedriagae* (Table 1).

### 3.3. Nuclear Genome of P. bedriagae

To assess introgression of the *P. bedriagae* nuclear genome (B) into *P. kl. esculentus* and *P. ridibundus*, we determined the sequence of the *Serum albumin* intron 1 (SAI-1) (Figure 4 and Table 1). Five *P. kl. esculentus* from P1 and P2, and five of the nine *P. ridibundus* from P2 and P3 possessed B haplotypes. One *kl. esculentus* female (P1–3) was a heterozygous BR, which might be a triploid with the mitochondrial *Cyt b* of the B haplotype (Table 1). None of the eight P4 *P. kl. esculentus* samples possessed any B haplotypes.

### 3.4. Genome Elimination in P. kl. esculentus

To verify if either genome is eliminated in the gonads of *P. kl. esculentus*, we compared the electrophoretic patterns of the SAI-1 fragments amplified from both finger and gonadal DNA. In 13 *P. kl. esculentus*, the bands of *lessonae* SAI-1 from the gonad DNA were clearly reduced in depth while those of *ridibundus* SAI-1 were slightly increased (Figure 5a,b; Table 1). This result suggests L genome elimination. In sharp contrast, in two *P. kl. esculentus* (P1–3 and P2–3), the depth of both species bands did not change between the finger and gonad DNA samples (Figure 5b, highlighted by a yellow box).

### 3.5. Genome Endoduplication in P. kl. esculentus

We aimed to verify whether the remaining genome (after genome elimination) is transferred to the progeny of *P. kl. esculentus*. Then, we crossed *P. kl. esculentus* with *P. ridibundus* or *P. lessonae* to determine the SAI-1 genotypes of the F1 offspring. The offspring of the *P. kl. esculentus* frogs were all RR homozygous when crossed with *P. ridibundus*, and were all LR heterozygous when crossed with *P. lessonae* (Appendix A). Therefore, the results show that the *P. kl. esculentus* all produced R sperm with the *P. ridibundus* genome. In sharp contrast, the four offspring of the *P. kl. esculentus* (P2–3) male crossed with *P. lessonae* female were of LL or LR genotype, demonstrating that the male produced both L and R sperm (Appendix A, left bottom).

### 3.6. Genome Elimination in the F1 Generation

To investigate the inheritance of the genome elimination system in *P. kl. esculentus*, we compared the SAI-1 patterns in the DNA from the fingers and gonads of the F1 offspring. We investigated the hybrid offspring from matings: between RR female and LL male, between LL female and RR male, and between LL female and the *P. kl. esculentus* male (P4–8). The results showed a clear decrease in L band depth in the gonads, suggesting L genome elimination (Appendix A, top). In contrast, the two F1 males from the mating between LL female and the *P. kl. esculentus* male (P2–3) showed no visible change in the R or L band depth between the finger and testis DNA samples (Appendix A, bottom and highlighted by a yellow box). This result suggests elimination of either species genome, similar to the male parent.

### 3.7. Hybridogenesis in Spermatogenic Cells

To investigate spermatogenesis in *P. kl. esculentus*, we histologically observed the inner structure of the testes in June (a breeding season). As expected by the lower fertility in some *P. kl. esculentus* frogs (Appendix A), a limited volume of sperm were observed in the restricted seminiferous tubules. Conversely, the control RR and LL frogs possessed seminiferous tubules filled with much sperm (Appendix A). Sperm were particularly scarce in the testes of the two *P. kl. esculentus* frogs (P2–2 and P2–3 in Appendix A). The low number of sperm and abnormal structure with fewer spermatogonia were also observed in *P. kl. esculentus* [26,27].

## 4. Discussion

### 4.1. Hybridogenesis in P. kl. esculentus

Hybridogenesis in the edible frog *Pelophylax kl. esculentus* is generally classified into two different systems of L-E and R-E. In the L-E system (in western and central Europe), the L genome is eliminated from gametogenic cells of *P. kl. esculentus*. The R gametes are produced through the endoduplication of the R genome and meiotic divisions [11,12,13,14]. On the other hand, in the R-E system, either genome can be eliminated and either or both gametes are produced in *P. kl. esculentus* [12,15,16,17,28]. An additional variation is the E–E system, where only *P. kl. esculentus* males and females live and either genome may be eliminated in both sexes. The RR and LL frogs produced from their mating may die out, with only *P. kl. esculentus* surviving, finally succeeding to a single lineage of *P. kl. esculentus* (in Switzerland) [29]. In addition, triploid LLR or RRL live with either parental species or *P. kl. esculentus*. This may eventually eliminate single or double genomes in the triploids or either genome in *P. kl. esculentus* (in Ukraine and Poland) [29,30,31,32]. Through the substantial volume of research, we have learned about the hybridogenetic mechanisms in *P. kl. esculentus.* This information includes the following findings: (1) genome elimination followed by endoduplication occurs during the proliferation of spermatogonia or oogonia before entering meiotic divisions [13,33,34,35], (2) determination of the eliminated genome depends on the sympatric parental species, and (3) these mechanisms do not depend on cytoplasmic origins [29,31,36,37]. Unfortunately, the most basic mechanisms remain a mystery.

### 4.2. Basic L-E System in Western Russian Territoty

In this study, we unveiled the taxonomic composition of the water frog complex in the populations of the western Russian territory and the reproductive mechanisms of *P. kl. esculentus*. Out of the four populations investigated, three (P1, P2 and P4) included *P. kl. esculentus*, while the other (P3) did not. The frogs in P4 had an L-E system, where *P. kl. esculentus* males live with *P. lessonae* males and females. The *P. kl. esculentus* spermatogenic cells eliminate the L genome, and R sperm were produced (Figure 6). The Y chromosome of *P. kl. esculentus* may have been carried by the clonal R genome. The mitochondrial *cytochrome b* of *P. kl. esculentus* possessed only haplotypes of *P. lessonae*. This L-E system under *lessonae* mitochondria is usually found in central Europe [36,38]. Therefore, this population may include descendants that have extended distribution east to Russia from the original European populations. In fact, we identified the haplotype of *Serum albumin* intron 1 of *P. ridibundus* that is specific to Poland and Germany in the genomes of *P. kl. esculentus* (P2–1 and P2–2; Figure 4).

### 4.3. Introgression from Marsh and Levant Frogs

On the other hand, the frogs from two other populations (P1 and P2) differed from P4 (Table 1). First, *P. kl. esculentus* males and females were coexisting in a population. Second, they lived with *P. lessonae* in P1 (as in P4), whereas they lived with both *P. lessonae* and *P. ridibundus* in P2. The P2 population can be called the L-R-E system, which has recently been identified in other populations in western Russian territory (Mari El Republic) [18]. Third, five of the seven *P. kl. esculentus* possessed the *cytochrome b* haplotypes of *P. ridibundus.* This is in sharp contrast to those of P4 who possessed *P. lessonae* haplotypes. Another most probable triploid individual (P1–3) in P1 had the *cytochrome b* haplotype of *P. bedriagae* and an individual from P2 (P2–3) possessed the *cytochrome b* haplotype of *P. lessonae* (Table 1). Finally, the most prominent differences were found in the *P. kl. esculentus* most probable triploid male (P1–1), which may possess no genome elimination, and in the *kl. esculentus* male (P2–3), which produced both R and L sperm. These are in sharp contrast to the *P. kl. esculentus* in P1, P2 and P4 who produced R (or B) gametes only (Figure 6).

We speculate that the following hybridogenetic mechanisms exist in the frogs of this western Russian territory: (1) L genome elimination basically does not depend on mitochondrial origins (L mitochondria in P4, and R or B mitochondria in P1 and P2 (except in one male)), and (2) the introduction of the marsh frog *P. ridibundus* and levant frog *P. bedriagae* into the original L-E system has contributed to the evolution of the L-R-E system (P2). This has produced both sexes of *P. kl. esculentus* by the introduction of X chromosomes of *P. ridibundus* or *P. bedriagae*, allowing RR survival, and (3) the introgression from *P. bedriagae* changed the hybridogenetic gametogenesis in one *kl. esculentus* male, from L genome elimination to L or R genome elimination under *lessonae* mitochondria (Figure 6). The third observation demonstrates a variation in the genome elimination within a population.

### 4.4. Variation in Genome Elimination

The only genetic difference detected between the *P. kl. esculentus* male (P2–3) which eliminated either the R or L genome and the others (P1–1,2 and P2–1,2,4) who eliminated the L genome only (in the P1 and P2 populations) is the *cytochrome b* haplotype. The *cytochrome b* haplotype was from *lessonae* in the P2–3 male, whereas it originated from *ridibundus* in the five *kl. esculentus* and was of *bedriagae* origin in the other *kl. esculentus* (P1–3, triploid). Thus, we hypothesise that the combined genome of R and B in *P. kl. esculentus* can be involved in L genome elimination under *ridibundus* mitochondria but not under *bedriagae* mitochondria (P1–3) (suggested by Plötner [39]). Furthermore, the combined R genome is eliminated under *lessonae* mitochondria, as observed in P2–3 (Figure 6). In fact, the offspring of the P2–3 *kl. esculentus* male mated with LL female also showed no change in the SAI-1 band depth between DNA samples from the finger and testis (Appendix A). This result suggests the elimination of either genome, similar to the parental male. In other words, the ancient, clonal *ridibundus* genomes in the L-E system are independently involved in L genome elimination under any *ridibundus* and *lessonae* mitochondria, as stated by Spolsky and Uzzell, Guerrini et al. and Ragghianti et al. [36,40,41]. If the R genomes are combined with the *bedriagae* genome, R genome elimination occurs under L mitochondria. The maternal involvement in genome elimination is also suggested in the hybrids of *Xenopus* species [42]. In the hybrids by a *X. tropicalis* female and *X. laevis* male, a few *laevis* chromosomes are lost from the cells during embryogenesis and the hybrid embryos finally die, but this does not occur in the opposite parental combination. Therefore, our conclusion is that recent introgression of the *P. bedriagae* genome into *P. kl. esculentus* resulted in mitochondria-dependent R genome elimination under L mitochondria. This has added to the original nuclear independent L genome elimination system, and thus provided the intrapopulation variation in genome elimination.

Selfish genetic elements are suggested to be involved in genome elimination of one parental species in hybridogenesis: they sweep in the R genome but not in the L genome, and then the L genome in which the elements have not spread is eliminated in the hybrids because the L genome has not evolved the corresponding suppressors [2,43,44]. This theory is applicable to the case in this study in which the B genome avoids the selfish elements and thus the combined R genome with B is eliminated in the hybrids under L mitochondria. The selfish genetic elements can be a species-specific signature on the genome to work on hybridogenesis and suggest the tendency of genome incompatibility to result in reproductive isolation between species. To identify the differences between the R and B genomes that may be responsible for the change in the genome elimination, whole genome analysis, including the search for transposable elements and suppressors, is expected.

## 5. Conclusions

We investigated hybridogenesis in the edible frog *Pelophylax kl. esculentus* from the populations in western Russian territory, which are located near the eastern end of its geographic distribution. Three out of four populations investigated included *P. kl. esculentus*, and the pool frog *P. lessonae* was predominant in the species composition. In addition, we identified introgression from another species, *P. bedriagae*, which is closely related to *P. ridibundus*, into the *ridibundus* genome in two populations. We confirmed in 13 *P. kl. esculentus* that the *lessonae* genome was eliminated and the R gametes were produced, which is an L-E system. In sharp contrast, in the testes of the other *P. kl. esculentus* male, the L or R genome was eliminated and R and L sperm were produced, which is an R-E system. Based on the genetic backgrounds of the *P. kl. esculentus* frogs, we hypothesized that the introgression from the levant frog *P. bedriagae* into the R genome was a trigger to produce variation in genome elimination, that is, a genetic change in choosing a genome to be eliminated, within a population.

## Figures and Tables

**Figure 1 genes-12-00244-f001:**
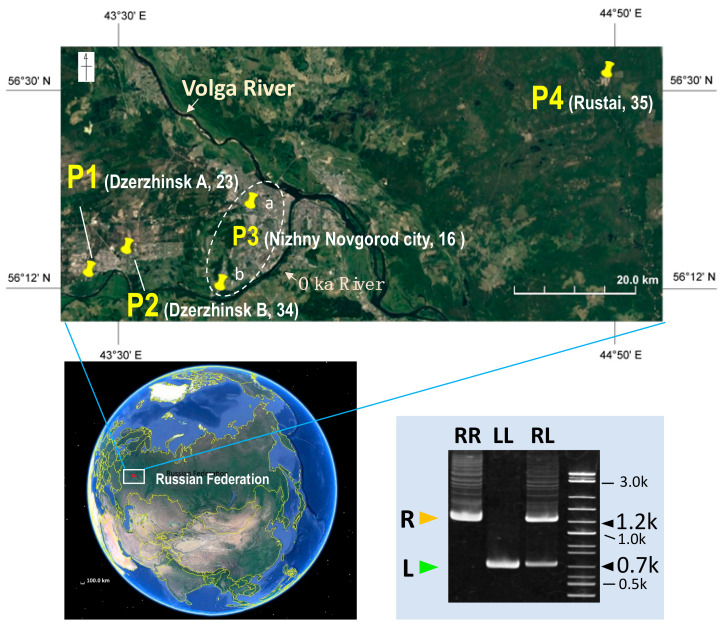
Maps showing the collecting locations of frogs and the taxonomic identification tool. The bottom left map indicates the Russian Federation on Earth, and the upper map is a magnification showing the five collection locations included in four populations of the frogs used in this study (the two locations a and b are combined to one population P3 because they are located within a city and either do not include any *P. kl. esculentus*). The total number of collected frogs in each population is described after the name of location in parenthesis. The right bottom picture is a taxonomic identification tool, showing an electrophoretic pattern of amplified fragments of *Serum albumin* intron 1 (SAI-1) in the three different frog species. RR, marsh frog *Pelophylax ridibundus*; LL, pool frog *P. lessonae*; RL, edible frog *P. kl. esculentus*. R, SAI-1 fragment of *P. ridibundus* and L, SAI-1 fragment of *P. lessonae*. The maps were created using the SAS-Planet package.

**Figure 2 genes-12-00244-f002:**
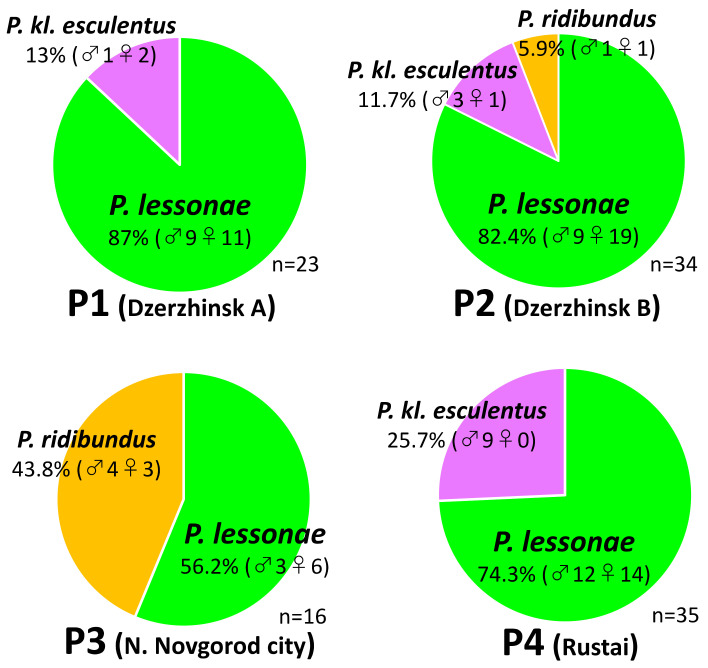
Taxonomic composition of *P. esculentus* complex in the sampled populations. The composition ratios of *P. lessonae*, *P. ridibundus* and *P. kl. esculentus* are demonstrated in green, yellow and purple, respectively. *P. kl. esculentus* was identified in P1, P2 and P4 but not in P3.

**Figure 3 genes-12-00244-f003:**
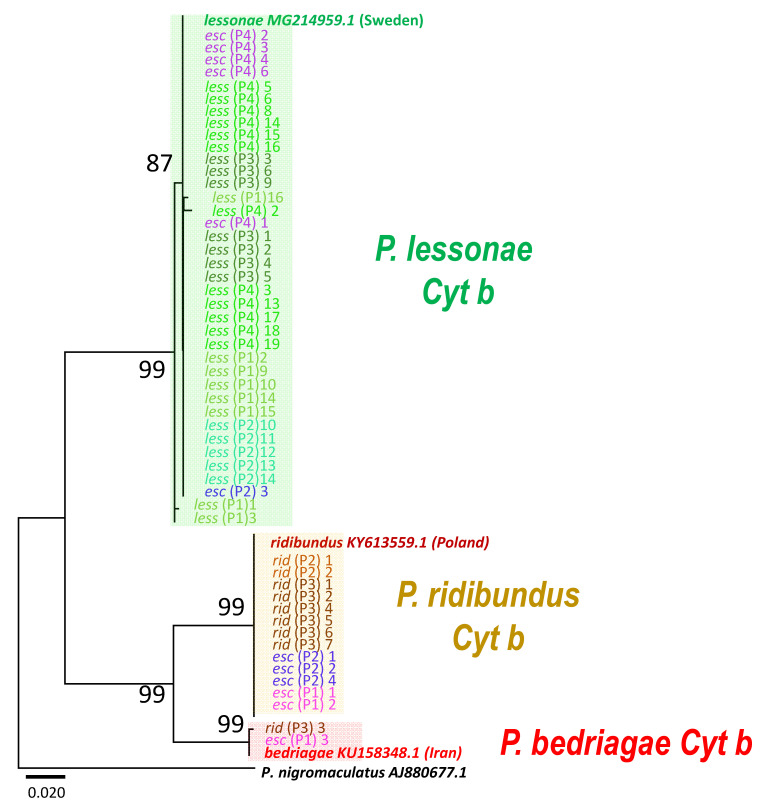
A mitochondrial *cytochrome b* tree showing the mitochondrial origins of *P. kl. esculentus*, constructed using the Maximum Likelihood method (ML). Three large clades of haplogroups of *P. lessonae*, *P. ridibundus* and *P. bedriagae* are highlighted by boxes in green, yellow and red, respectively. The *cytochrome b* haplotypes of *P. kl. esculentus* from P4 were all included in the *lessonae* clade (green box), while those of *P. kl. esculentus* from P1 and P2 belonged to the *lessonae, ridibundus* or *bedriagae* clades. One *ridibundus* (P3–3) possessed the haplotype of *P. bedriagae*. The numbers at the nodes of the tree are the percentages of 500 bootstrap replicates.

**Figure 4 genes-12-00244-f004:**
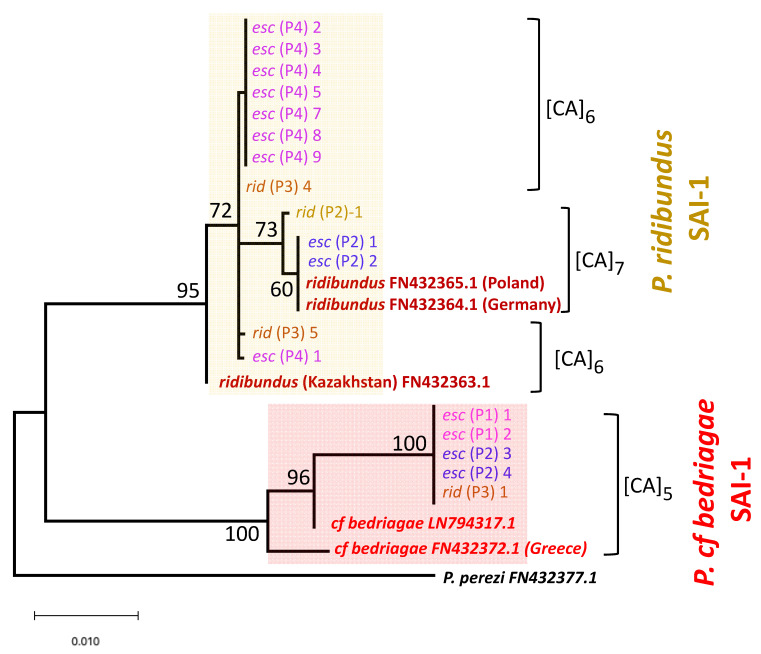
Nuclear *Serum albumin* intron 1 tree in *P. kl. esculentus* and *P. ridibundus*, constructed using the Maximum Likelihood method. The two large clades of *P. ridibundus* and *P. bedriagae* are highlighted using a box of yellow and red, respectively. The SAI-1 haplotypes of *P. kl. esculentus* from P4 all belonged to the *ridibundus* clade, while those of *P. kl. esculentus* from P1 and P2 belonged to either clade. [CA]_n_ indicates the CA repeat (n, number of repeats) located at the 3′ flanking region of the transposon included in the *ridibundus* and *bedriagae* intron 1. The numbers at the nodes of the tree are the percentages of 500 bootstrap replicates.

**Figure 5 genes-12-00244-f005:**
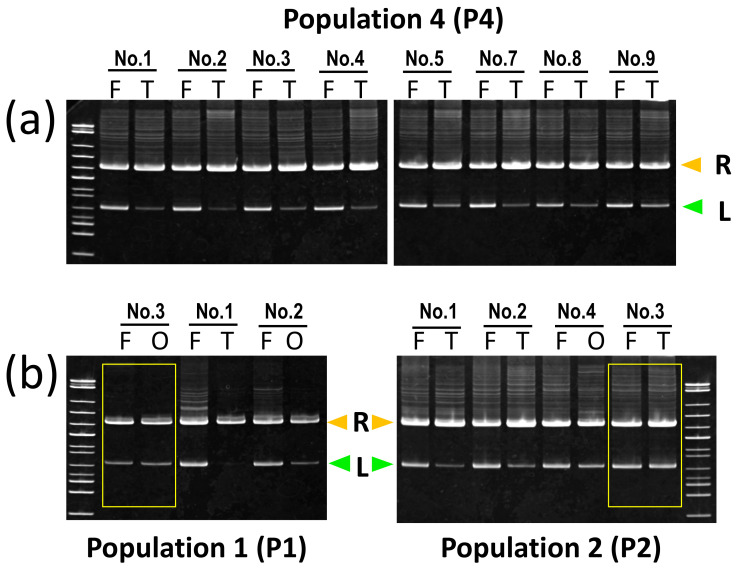
Genome elimination in *P. kl. esculentus.* SAI-1 fragments amplified from finger (F) and gonad DNA (T, testis; O, ovary) in eight *P. kl. esculentus* from P4, three from P1 and four from P3 are shown. The L bands (*lessonae*) of these samples from gonads are all much weaker in depth while R bands (*ridibundus*) are slightly increased compared to those from the finger DNA, indicating L genome elimination in the gonads (**a**,**b**). On the other hand, one female and one male *P. kl. esculentus* from P1 and P2, respectively, did not show any difference in the depth of the SAI-1 bands between the finger and gonad DNA (**b**, boxed in yellow).

**Figure 6 genes-12-00244-f006:**
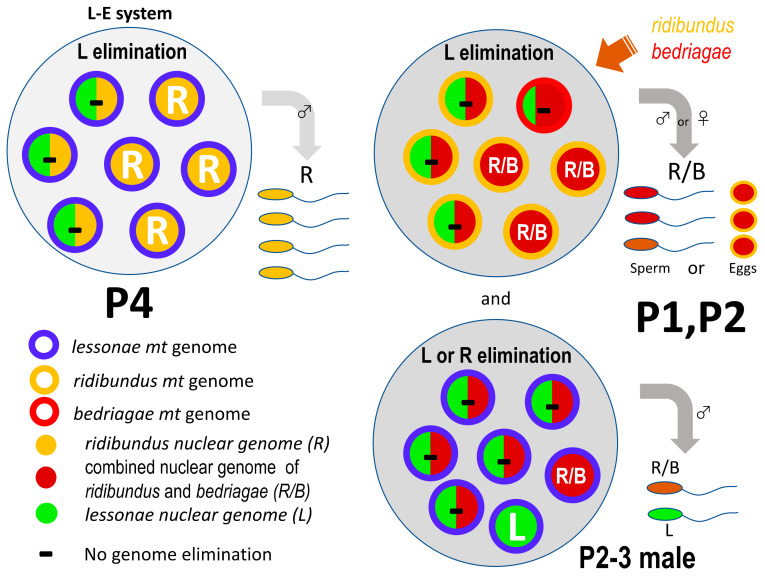
Diagram showing presumed hybridogenetic gametogenesis in *P. kl. esculentus* from the western Russian territory. P4 possesses an L-E system, where the L genome is eliminated in the testes of *P. kl. esculentus* and R sperm are produced under L mitochondria. Likewise, P1 and P2 had an L-E system, where the L genome is eliminated in the gonads and R/B gametes (combined genome of *ridibundus* and *bedriagae*) are produced only under the *ridibundus* mitochondria. In contrast, neither genome might be eliminated in one *kl. esculents* triploid female (P1–3) under *bedriagae* mitochondria. Furthermore, in the one *kl. esculentus* (P2–3) individual, the L or R genome is eliminated in the testis and the R and L sperm are produced under *lessonae* mitochondria. This represents genome elimination in the R-E system. Large circles indicate gonads of *P. kl. esculentus*, in which small circles are germ cells: mitochondria are shown by color of outline and nuclear genomes are shown by inner color or letter. Abbreviations are shown in the figure.

**Table 1 genes-12-00244-t001:** Mitochondrial and nuclear haplotypes or genotypes and hybridogenesis in the water frog complex from Russian territory.

Species	Population	Individual Number	Sex	*Cyt b ^(^* ^1)^	*Serum A* Intron 1 (SAI-1) ^(2)^	Reduced SAI-1 Allele in Gonad ^(3)^	SAI-1 Allele of Gamete	Fertility ^(4)^
***P. esculentus***	P1	No.1	male	R	B	L	NE	
		No.2	female	R	B	L	NE	
		No.3	female	B	B R	No change	NE	
	P2	No.1	male	R	R	L	R	42.8
		No.2	male	R	R	L	NE	0
		No.3	male	L	B	No change	B or L	0–2.3
		No.4	female	R	B	L	B	8.5
	P4	No.1	male	L	R	L	R	66.7
		No.2	male	L	R	L	NE	
		No.3	male	L	R	L	R	37.5
		No.4	male	L	R	L	R	63
		No.5	male	L	R	L	R	11.3
		No.6	male	L	NE	NE	NE	
		No.7	male	L	R	L	R	44.4
		No.8	male	L	R	L	R	36.4–50.9
		No.9	male	L	R	L	R	1.5–33.3
***P. ridibundus***	P2	No.1	male	R	RR			
		No.2	female	R	R B			
	P3	No.1	male	R	BB			
		No.2	male	R	RR			30.4–75.1
		No.3	male	B	R B			
		No.4	male	R	RR			
		No.5	female	R	RR			
		No.6	female	R	R B			75.1

NE, not examined. ^(1)^ Haplotype of mitochondrial *cytochrome b.*
^(2)^ Haplotype or genotype of *Serum albumin* intron 1. ^(3)^ The letter of species genome, of which serum albumin intron 1 fragment band depth is reduced in gonadal DNA compared to that of finger DNA; ^(4)^ Fertility is the ratio of normal tail-bud embryos to the total number of layed eggs (cf. Appendix A). Green refers to *P. lessonae*, yellow refers to *P. ridibundus*, red refers to *P. bedriagae*.

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
