# Peer review of "Hybridogenesis in the Water Frogs from Western Russian Territory: Intrapopulation Variation in Genome Elimination"

_genes, 2021, doi:10.3390/genes12020244_

Round 1
Reviewer 1 Report
Brief summary
The manuscript entitled "Hybridogenesis in the water frogs from Russian populations: Transition in genome elimination system" by Miura et al. is describing breeding system of water frogs in the Pelophylax esculentus complex in Russia with an interesting observation of P. esculentus hybrid producing sperms with both parental genomes. The authors hypothesize that this could be a transition between the LE breeding system to RE system in one particular population. This finding based on one observation (i.e., a single hybrid individual) which makes this assumption to highly questionable.
Broad comments
First, the introduction of the study is rather superficial, lacking basic information about the breeding system diversity of water frogs, the distribution of P. bedrigiae and their capacity the get involved in hybridogenetic breeding systems. These gaps are needed to be filled.
However, my major issue is the presentation of the results. The quality of the figures and tables are not sufficient to appropriately present the results. In some places the results were detailed in the text were not in line with their presentation in the figures (e.g., Figure 5). Basically, all the figures and tables are unnecessarily 'colorful' with inconsistent usage of the different colours which makes the main findings hard to follow for the reader. Furthermore, some major results are presented as a Supplementary Material in which also hard to find it. The main text and the supplement is full with typos and the gel photos with the subtitles are seems chaotic and not possible to follow.
Overall, the main findings of the manuscript (i.e., the taxon composition and the observed breeding system of Russian populations) is worth to publish because it adds information to the breeding system diversity of P. esculentus complex but I will be more cautious with the transition part between breeding systems observing only one individual acting differently than others in the population.
Specific comments
Introduction
Lines 40-41: This argument should be explained more in detail and references should be provided.
Line 49: Use edible frog instead of European water frog as a vernacular name for Pelophylax esculentus.
Line 49: "(E or RL)" Use only the RL genotype throughout the whole text and the supplement as well. Moreover, after the first mention the authors should indicate these are genotypes and use them consistently.
Lines 49-56: In its current form this paragraph is incomplete. At least in a few sentences the authors should mention the remaining breeding systems i.e., the all hybrid (EE) system where triploid hybrids are taking over the role of parental species in maintaining the population. Furthermore, the L-E-R population should be detailed where both parental species and the hybrids are living syntopically.
Lines 55-56: The authors need to specify the aims of this study. If these are belonging to the aims, please indicate it and unify.
Lines 57-58: The authors mean that these are newly discovered populations of P. esculentus complex 800 km towards to East from the easternmost known population?
Lines 58-59: "To date, few genetic studies have included the Russian populations." References? In addition, it would be nice to see a short overview of these studies with the key findings.
Figure 1: The resolution of Figure 1 is far from acceptable. I advise that to exclude the whole globe view and focus on study sites instead. More specifically, raster maps would be more welcome. Moreover, it would be informative if the number of collected individuals per sampling sites (P1-4) will be included to the map or at least to the figure legend.
In addition, serum albumin intron 1 should not be in italic and should be abbreviated to SAI-1 through the whole text.
In fine, "species identification tool" needed to be separated to a single figure or what could be more appropriate is to detail it in the species identification part of the M&M. i.e., detail the core of the method, the fragment lengths and specificity of each members of the complex. Finally, we do not know anything about the scale of the DNA ladder used therefore the taxa specific fragment lengths.
Material and Methods
Line 80: Indicate the method/substance you used for euthanasia. I guess after that the individuals were dissected. Add this information to the text.
Line 94: You mean EmeraldAmp MAX PCR Master Mix?
Results
Line 110: It is more appropriate to use taxonomic composition instead of species composition. Namely, P. esculentus is a hybridogenetic hybrid i.e. a klepton which sexually parasitize its parental species as hosts. I would beware to call it a real species as P. ridibundus and P. lessonae; however, there are all-hybrid populations in which sexual reproduction successfully takes place without the presence of the parental species. There are several papers dealing with this breeding system and the gamete production of hybrids. In fine, I would recommend to use the term 'klepton' or 'hybridogenetic hybrid species' instead of 'species' when you refer to P. esculentus in the manuscript.
Line 115: "(56.2 ~ 87 %)". Indicate that this is a range of P.lessonae occurrance in the sample and do the same for the other two members of the complex.
Figure 2: These pie charts would be more eye-catchy without colours and simply represented in B&W (just think about colour blind readers). In addition, the name of three taxa should be with the same font size and colour. Not consistent colouring is confusing and misleading and this general impression of mine after checking all the figures is not changed at all.
Lines 125-126: What is the source of the additional sequences from Sweden, Greece, Poland...etc? I think these were used for reference to lessonae, ridibundus and bedriagae haplotypes. Anyway, add background information to the M&M section about these and for the outgroups used to construct the phylogenetic trees as well.
Figure 3: "P. nigromaculata AJ880677". I guess this is a GenBank accession number, but I cannot found anything related about the outgroups used for any of the analyses presented in this manuscript.
Again. Inconsistent colouring is very confusing. Colours are not coded adequately and not refer the same species in every figures including supplementary materials. Sometimes simple is better.
Table 1: Formally this table is unacceptable in highly-ranked journals like Genes. Abbreviations used in the table is not explained. The same is true for tables and figures in the supplement. Moreover, how did the authors quantified fertility in hybrids? There is a column with "Fertility" but I did not find information in the text about the meaning of numbers in each cell...
Figure 4: What is the authors explanation for the low (=3) bootstrap value in esculentus individuals grouped together in the P4 population?
Line 146. How did the authors examined the ploidy level of the individual (P1-3 female) that presenting as a triploid? To define ploidy level with the highest confidence is the application of microsatellite markers complemented with erythrocyte measurements. Did the authors do that?
Line 158: There are 15 samples in Figure 5. but the authors mention 14 in the text.
Figure 5: What does the F T and F O means? Finger and testis/ovary? Moreover, the authors stated that in the figure legend "SAi1 fragments amplified from finger and gonad DNA 164 in all P. esculentus from P4, two from P1 and three from P3 are shown" in contrast in the figure I can see 8 samples from P4; 3 samples from P1 and 4 from P2. Finally, this method is not adequate at all to confirm genome elimination only good for to check the taxa-specific bands represented in the gel. The cause of the band intensity differences between the extractions of fingers and gonads are influenced by several factors e.g., initial DNA content or contamination.
Author Response
#The line numbers in comments are those of the original text, while the lines in our responses are those of the revised text.
Reviewer 1
Thank you for the comments.
Our responses to the comments are written in purple.
Brief summary
The manuscript entitled "Hybridogenesis in the water frogs from Russian populations: Transition in genome elimination system" by Miura et al. is describing breeding system of water frogs in the Pelophylax esculentus complex in Russia with an interesting observation of P. esculentus hybrid producing sperms with both parental genomes. The authors hypothesize that this could be a transition between the LE breeding system to RE system in one particular population. This finding based on one observation (i.e., a single hybrid individual) which makes this assumption to highly questionable.
Broad comments
First, the introduction of the study is rather superficial, lacking basic information about the breeding system diversity of water frogs, the distribution of P. bedrigiae and their capacity the get involved in hybridogenetic breeding systems. These gaps are needed to be filled.
Response:
The basic information about hybridogenesis is included in the introduction and also in discussion. If adding more information, it would be confused because the variations in this system is huge and very complicated. we think the descriptions in introduction and in the first paragraph of discussion are enough for readers to catch the general phenomena of hybridogenesis and discuss about the results of this study.
On the other hand, we added information about P. bedriagae in Russian territory with the references in the introduction. Please see the revised introduction.
However, my major issue is the presentation of the results. The quality of the figures and tables are not sufficient to appropriately present the results. In some places the results were detailed in the text were not in line with their presentation in the figures (e.g., Figure 5). Basically, all the figures and tables are unnecessarily ‘colorful’ with inconsistent usage of the different colours which makes the main findings hard to follow for the reader. Furthermore, some major results are presented as a Supplementary Material in which also hard to find it. The main text and the supplement is full with typos and the gel photos with the subtitles are seems chaotic and not possible to follow.
Response:
Colours:
Based on the comment, we unified the usage of colours in figures and tables to identify each species more easily: green for lessonae, yellow or brown for ridibundus, purple or pink for esculentus and red for bedriagae.
Supplemental figures:
we added a large title to make it easier to catch the results.
Overall, the main findings of the manuscript (i.e., the taxon composition and the observed breeding system of Russian populations) is worth to publish because it adds information to the breeding system diversity of P. esculentus complex but I will be more cautious with the transition part between breeding systems observing only one individual acting differently than others in the population.
Response:
This underlined comment was given by another reviewer. It is just one male who eliminated L or R genome in the testes. However, the result is true. Please see the figure S3 bottom. It is evident that two offspring of the esculentus male mated with LL female are LL and the other two are RL. In addition, we investigated the DNA of fingers and testes in the two RL sons and got the same result as the father, suggesting L or R elimination. It is necessary to investigate more to confirm this variation and thus we are bearing the two sons in order to cross them and do genome analysis. If having a chance, we will also challenge investigation to collect more esculentus from the same P2 population.
Specific comments
Introduction
Lines 40-41: This argument should be explained more in detail and references should be provided.
We added one reference in L43:
“However, the nature of “genomic incompatibility” remains unclear (Burt and Trivers, 2006).”
Please suggest us references to add, if you have more.
Line 49: Use edible frog instead of European water frog as a vernacular name for Pelophylax esculentus.
L51:
European water frog” was revised to !edible frog”.
Line 49: "(E or RL)" Use only the RL genotype throughout the whole text and the supplement as well. Moreover, after the first mention the authors should indicate these are genotypes and use them consistently.
We unified the usage of RL throughout the text, figures and supplemental figures.
Lines 49-56: In its current form this paragraph is incomplete. At least in a few sentences the authors should mention the remaining breeding systems i.e., the all hybrid (EE) system where triploid hybrids are taking over the role of parental species in maintaining the population. Furthermore, the L-E-R population should be detailed where both parental species and the hybrids are living syntopically.
Two other breeding systems are introduced in the first paragraph of discussion. As mentioned in our response to the first comment above, the variations in hybridogenesis are complicated and thus we avoid the detailed description particularly in introduction instead the basic systems only.
Lines 55-56: The authors need to specify the aims of this study. If these are belonging to the aims, please indicate it and unify.
Our aim is added in the head of next paragraph in L59 as follows:
“Are the two systems stable or changeable with each other? What genetic factor is involved in choosing the genome to be eliminated?
In this study, to approach the above questions, we investigated the water frogs from western Russian territory, populations of which are located 800 km west from the eastern end of their geographic distribution (Figure 1).
Lines 57-58: The authors mean that these are newly discovered populations of P. esculentus complex 800 km towards to East from the easternmost known population?
Not new populations. It means that the populations we investigated are located 800 km west from the eastern end of known geographic distribution of the water frogs. So, we added “west” in the text in L60.
Lines 58-59: "To date, few genetic studies have included the Russian populations."References? In addition, it would be nice to see a short overview of these studies with the key findings.
We added the references as follows in L61-64:
“To date, few genetic studies have included the populations (Dedukh et al, 2019) and unusual situation of hybridogenesis in the water frog complex is expected to be brought about by immigration of another levant frog into the Russian territories (Svinin et al., 2015; Ivanov et al., 2015; Zamaletdinov et al., 2015). Our study identified a variation in the genome elimination within one population.”
Figure 1: The resolution of Figure 1 is far from acceptable. I advise that to exclude the whole globe view and focus on study sites instead. More specifically, raster maps would be more welcome. Moreover, it would be informative if the number of collected individuals per sampling sites (P1-4) will be included to the map or at least to the figure legend.
The resolution of figure 1 was up to 300 dpi.
Global map is necessary because most people in the world are not familiar to the western Russian territory we investigated.
We added the total number of frogs collected, the names of rivers and coordinates on the map.
In addition, serum albumin intron 1 should not be in italic and should be abbreviated to SAI-1 through the whole text.
We revised all SA1-1from italics to roman.
In fine, "species identification tool" needed to be separated to a single figure or what could be more appropriate is to detail it in the species identification part of the M&M. i.e., detail the core of the method, the fragment lengths and specificity of each members of the complex. Finally, we do not know anything about the scale of the DNA ladder used therefore the taxa specific fragment lengths.
The electrophoretic picture should be included in either of figures but should not be included in supplemental figure because it is important. So, we placed it in figure 1.
The fragment size of SAI-1 is put on the picture: R band of SAI-1 is 1.2kb and L band of SAI-1 is 0.7kb as shown there. The precise length of SAI-1 is different in every fragments because it is an intron including deletion and insertion and the number of CA repeats depends on individuals. Please see the details in the figure 3 for CA repeat length.
We added size of markers in figure 1 and also information of the DNA ladder marker we used was added in the materials and methods as follows in L118-121:
For genotyping of SAI-1, the amplified product was electrophoresed in 6% acrylamide gel with 1 x TBE (Tris, Boric Acid and 2Na-EDTA) buffer with DNA ladder marker (Quick-load purple 1kb plus DNA ladder, BioLabs), and was photographed using UV illuminator after stained with ethidium bromide.
Material and Methods
Line 80: Indicate the method/substance you used for euthanasia. I guess after that the individuals were dissected. Add this information to the text.
We used 0.1−1.0% MS222, which is added in the Materials and methods as follows in L88-90:
“The sex of the specimens was determined based on the external morphology, such as black thumb callus of males and eggs of females, or by inspection of gonads after euthanasia (soaked in 0.1 – 1.0% MS222).”
Line 94: You mean EmeraldAmp MAX PCR Master Mix?
Thanks. It was revised to EmeraldAmp PCR master mix (TaKaRa, Japan)
Results
Line 110: It is more appropriate to use taxonomic composition instead of species composition. Namely, P. esculentus is a hybridogenetic hybrid i.e. a klepton which sexually parasitize its parental species as hosts. I would beware to call it a real species as P. ridibundus and P. lessonae; however, there are all-hybrid populations in which sexual reproduction successfully takes place without the presence of the parental species. There are several papers dealing with this breeding system and the gamete production of hybrids. In fine, I would recommend to use the term 'klepton' or 'hybridogenetic hybrid species' instead of 'species' when you refer to P. esculentus in the manuscript.
“Species composition” was changed to “taxonomic composition” in the text.
And, P. esculentus were all revised to P. kl. esculentus.
Line 115: "(56.2 ~ 87 %)". Indicate that this is a range of P.lessonae occurrance in the sample and do the same for the other two members of the complex.
The information about P. ridibundus was added in L129. The rate of P. esculentus is described in the next sentence in L129-130:
“In all four populations, P. lessonae was predominant in the constitution (56.2 ~ 87 %, while P. ridibundus, 0 ~ 43.8%). P. kl. esculentus was identified in the three populations of P1, P2 and P4 (11.7 ~ 25. 7%).”
Figure 2: These pie charts would be more eye-catchy without colours and simply represented in B&W (just think about colour blind readers). In addition, the name of three taxa should be with the same font size and colour. Not consistent colouring is confusing and misleading and this general impression of mine after checking all the figures is not changed at all.
As stated above, the color usage was unified based on species in text, figures and tables. The color of species name was all changed to black according to the comment. On the other hand, we want to stay the size of letter as they were, because the larger letter of lessonae species helps to indicate that the species is predominant in composition.
Lines 125-126: What is the source of the additional sequences from Sweden, Greece, Poland...etc? I think these were used for reference to lessonae, ridibundus and bedriagae haplotypes. Anyway, add background information to the M&M section about these and for the outgroups used to construct the phylogenetic trees as well.
The informations were added to Materials and methods in L114-118 as follows:
“Accession Nos. LC599369~LC599378 for cytochrome b and LC599379~LC599385 for Serum albumin intron 1. The sequence data of cytochrome b (MG214959.1, KY613559.1, KU158348.1 and AJ880677.1) and Serum albunin intron 1 (FN432363.1, FN432364.1, FN432365.1, FN432372.1, LN794317.1 and FN432377.1) were obtained from GenBank.”
Figure 3: "P. nigromaculata AJ880677". I guess this is a GenBank accession number, but I cannot found anything related about the outgroups used for any of the analyses presented in this manuscript.
They are all deposited on the bank and we cited them from there. Please check again in NCBI. Outgroups we used are P. nigromaculatus (nigromatulata was a misspelling) in figure 2 and P. perezi in figure 3.
Again. Inconsistent colouring is very confusing. Colours are not coded adequately and not refer the same species in every figures including supplementary materials. Sometimes simple is better.
Thank you for this comment. We unified the color usage based on species in figures and tables as stated above.
Table 1: Formally this table is unacceptable in highly-ranked journals like Genes. Abbreviations used in the table is not explained. The same is true for tables and figures in the supplement. Moreover, how did the authors quantified fertility in hybrids? There is a column with "Fertility"but I did not find information in the text about the meaning of numbers in each cell...
Explanation of each column was added in table 1 as follows:
1) Haplotype of mitochondrial cytochrome b
2) Haplotype or genotype of serum albumin intron 1
3) The letter of species genome, of which serum albumin intron 1 fragment band depth is reduced in gonadal DNA compared to that of finger DNA
4) Fertility is the rate of normal tail-bud embryos to the total number of layed eggs (cf. Table S2)
Figure 4: What is the authors explanation for the low (=3) bootstrap value in esculentus individuals grouped together in the P4 population?
It just indicates that the sequence difference between the P4 esculentus sub-clade and others in the ridibundus clade is very low. Thus, “3” was removed from the figure.
Line 146. How did the authors examined the ploidy level of the individual (P1-3 female) that presenting as a triploid? To define ploidy level with the highest confidence is the application of microsatellite markers complemented with erythrocyte measurements. Did the authors do that?
No we didn’t do it. it is based on the heterozygosity of SAI-1 for the R genome of the esculentus frog. We have no ideas to explain the heterozygosity other than triploidy. In addition, the relative difference of band depth between R and L bands of SAI-1 in the frog is different from those in the other RL: L band is much fainter than R band (please see figure 5-b). The frog was transported to our lab after death. So, we could not check the karyotypes. We examined others, which were all diploids. Because we have no other data to confirm its triploidy, “may” was changed to “might” in L 159 as follows:
“One kl. esculentus female (P1-3) was a heterozygous BR, which might be a triploid with the mitochondrial Cyt b of the B haplotype (Table 1).”
Line 158: There are 15 samples in Figure 5. but the authors mention 14 in the text.
This is our mistake. It is 13 instead of 14 described in the text and thus was revised in L171. L band is weaker in 13 RL, while is not different from R band in the other two. Total number investigated is 15.
Figure 5: What does the F T and F O means? Finger and testis/ovary? Moreover, the authors stated that in the figure legend "SAi1 fragments amplified from finger and gonad DNA in all P. esculentus from P4, two from P1 and three from P3 are shown" in contrast in the figure I can see 8 samples from P4; 3 samples from P1 and 4 from P2. Finally, this method is not adequate at all to confirm genome elimination only good for to check the taxa-specific bands represented in the gel. The cause of the band intensity differences between the extractions of fingers and gonads are influenced by several factors e.g., initial DNA content or contamination
“F, finger and T, testis; O, ovary” were added in the figure 5 legend as follows:
“Figure 5. Genome elimination in P. kl. esculentus. SAI-1 fragments amplified from finger (F) and gonad DNA (T, testis; O, ovary) in eight P. kl. esculentus from P4, three from P1 and four from P3 are shown.”
The numbers were revised to eight from P4, three from P1 and four from P2 as above.
They were our mistakes.
About the accuracy of the SAI-1electrophoretic method to see the genome elimination:
We are sure that it is an excellent and simple method to know which genome is eliminated. What should be compared from each other is the band depth between R and L bands from the same tissue and then the patterns between the finger and gonad. The quality of DNA or initial DNA content does not affect the result. Please see the results of figure 5. L bands from gonadal DNA are clearly reduced in 13 of 15 esculentus relative to R bands, while the depth of R bands is not so different in every samples; similar depth of R bands are well seen commonly between finger and gonad in the same individual.
In addition, to confirm the eliminated genome, we crossed the esculentus with LL or RR and examined the genotypes of SAI-1 of the offspring. We also confirmed the results of SAI-1 analyses in finger and gonadal DNA by seeing the wave height of base at the heterozygous site of SOX3. This also well supported the results of SA1-A analysis.
The most important thing for this SAI-1 analysis is the primers for amplification. The conventional ones published in previous paper do not work for this, thus we newly designed the present primers to amplify equally the SAI-1 fragments from R and L genomes .
Reviewer 2 Report
Miura et al. proposes a description of the hybridogenetic system observed in four Russian populations. They documented cases in which one or both parental species coexist in the system, as well as cases in which introgression from a third species is also observed. They propose that introgression from this third species may participate in the transition from one parental genome elimination to another. While the main idea and analysis of the manuscript are useful and suitable for publication, there are some concerns that I briefly describe below:
- The title, abstract and several sections of the manuscript give the impression you are characterising a broad range of Russian populations. However, the four locations seem to be distanced by less than 100kms, with three of them separated by around 20km. These different locations are not necessarily independent. You may include a statement specifying the reason these locations are considered to be different populations. In addition, you may clearly say that these results are valid for these four locations only, not for the wide range of Russian populations.
- The manuscript starts with the biological species concept, but the influences of this concept in the definition of species for these frogs is not developed in the manuscript, which make this starting a bit "out of topic". You may consider saying something about these later in the discussion or modify this paragraph. The same paragraph also finishes with a statement about the poor knowledge regarding the "genomic incompatibility" between divergent lineages, but this is also not the main problem explored in the manuscript.
- The main objective or novelty of this manuscript also has to be presented more clearly in the introduction. Some questions are stated in lines 55-56, but neither of them represents questions fully answered by this manuscript. You may consider modifying them or replacing them with the main objectives (or questions) of this manuscript. It is better to include this at the end of the last paragraph of the introduction.
- In the discussion, you also start with an unanswered question regarding the essential factor causing a genome to be eliminated. While more research regarding a specific study system is still needed, there is broad literature about “selfish genetic elements” that can be considered to answer this kind of question. There is a section about selfish genetic elements and hybridogenesis in Quilodran et al. (2018) Conservation Biology.
Minor comments:
- Line 21. Could you be more specific about the “essential mechanisms” that remain unclear?
- Line 21. You may consider saying: "Here, we focussed on four Russian populations"
- Line 42. Add a reference at the end of the first sentence.
- Line 55. "Decide" is probably not the best word to used here.
- Lines 59-60. Could you delete or modify the sentence: "Therefore, we anticipate new findings in these obscure populations"? In the previous sentence, you have already said that these populations have been poorly studied, which is already sufficient justification for your study.
- Figure 1. Add a geographical coordinate reference for the top figure.
- Line 112. Please use italics for all scientific names. Check all along the manuscript (e.g., line 116).
- Figure 3. The font size of the number “0.020” at the bottom left of the figure is too small.
- Line 214 and Line 248. Specify that you are considering "four" Russian populations. However, define earlier in the manuscript if these locations can actually be considered different populations.
- Line 293. "The following figures and tables"?
- You may consider asking a native English speaker to read the full manuscript.
Author Response
#The line numbers in comments are those of the original text, while the lines in our responses are those of the revised text.
Reviewer 2
Thank you for the comments. Our responses are written in blue.
Comments and Suggestions for Authors
Miura et al. proposes a description of the hybridogenetic system observed in four Russian populations. They documented cases in which one or both parental species coexist in the system, as well as cases in which introgression from a third species is also observed. They propose that introgression from this third species may participate in the transition from one parental genome elimination to another. While the main idea and analysis of the manuscript are useful and suitable for publication, there are some concerns that I briefly describe below:
- The title, abstract and several sections of the manuscript give the impression you are characterising a broad range of Russian populations. However, the four locations seem to be distanced by less than 100kms, with three of them separated by around 20km. These different locations are not necessarily independent. You may include a statement specifying the reason these locations are considered to be different populations. In addition, you may clearly say that these results are valid for these four locations only, not for the wide range of Russian populations.
We changed the description from “Russian populations” to “western Russian territory” in title.
The definition of population is based on the taxonomic (species) composition, and locations 3a and 3b are united to one population because they are located in a city and either do not include any P. kl. esculentus. This description was added in the materials and methods and figure 1 legend as follows:
In M. & M., L78-:
“108 adult frogs in total were collected from five locations in the middle of East European plane of Russia in May to September during 2017 to 2019 (Figure 1). The five locations are classified into four different populations (P1 ~ P4) based on the taxonomic composition: the two locations in P3 population are distant from each other but are combined to one population because the two are located within a city and either do not include any P. kl. esculentus..”
In figure 1 legend in L67-70:
“, and the upper map is a magnification showing the five collection locations included in four populations of the frogs used in this study (The two locations a and b are combined to one population P3 because they are located within a city and either do not include any P. kl. esculentus).”
Title:
“Hybridogenesis in the water frogs from western Russian territory: Intrapopulation variation in genome elimination”
“Abstract: Hybridogenesis in an interspecific hybrid frog is a coupling mechanism in the gametogenic cell line that eliminates the genome of one parental species with endoduplication of the remaining parental species. It has been intensively investigated in the water frog Pelophylax kl. esculentus (RL), a natural hybrid between the marsh frog P. ridibundus (RR) and the pool frog P. lessonae (LL). However, the genetic mechanisms involved remain unclear. Here, we investigated the water frogs in the western Russian territory. In three of the four populations, we genetically identified 16 RL frogs living sympatrically with the parental LL species, or with both parental species. In addition, two populations contained genome introgression with another species, P. bedriagae (BB) (a close relative of RR). In the gonads of 13 RL frogs, the L genome was eliminated, producing gametes of R (or R combined with the B genome). In sharp contrast, one RL male eliminated the L or R genome in the testes, producing both R and L sperm. We detected a variation in genome elimination within a population. Based on the genetic backgrounds of RL frogs, we suggest that the introgression of the B genome resulted in the change in choosing a genome to be eliminated.”
- The manuscript starts with the biological species concept, but the influences of this concept in the definition of species for these frogs is not developed in the manuscript, which make this starting a bit "out of topic". You may consider saying something about these later in the discussion or modify this paragraph. The same paragraph also finishes with a statement about the poor knowledge regarding the "genomic incompatibility" between divergent lineages, but this is also not the main problem explored in the manuscript.
We added in the final paragraph a discussion about “selfish genetic elements” in relation to the mechanism of genome elimination as follows in L300-309:
“Selfish genetic elements are suggested to be involved in genome elimination of one parental species in hybridogenesis: they sweep in R genome but not in L genome, and then L genome in which the elements have not spread is eliminated in the hybrids because L genome has not evolved the corresponding suppressors [2, 43, 44]. This theory is applicable to the case in this study that B genome avoids the selfish elements and thus the combined R genome with B is eliminated in the hybrids under L mitochondria. The selfish genetic elements can be a species specific signature on genome to work on hybridogenesis and suggest the nature of genome incompatibility to result in reproductive isolation between species. To identify the differences between the R and B genomes that may be responsible for the change in the genome elimination, whole genome analysis with searching transporsable elements and the suppressors is expected.
- The main objective or novelty of this manuscript also has to be presented more clearly in the introduction. Some questions are stated in lines 55-56, but neither of them represents questions fully answered by this manuscript. You may consider modifying them or replacing them with the main objectives (or questions) of this manuscript. It is better to include this at the end of the last paragraph of the introduction.
Thanks for the comments.
We rewrote the last paragraph of introduction in L59-64:
“In this study, to approach the above questions, we investigated the water frogs from western Russian territory, of which populations are located 800 km west from the eastern end of their geographic distribution (Figure 1). To date, few genetic studies have included the populations [18] and unusual situation of hybridogenesis in the water frog complex is expected to be brought about by immigration of another levant frog into the Russian territories [19−21]. Our study identified a variation in the genome elimination within one population.”
- In the discussion, you also start with an unanswered question regarding the essential factor causing a genome to be eliminated. While more research regarding a specific study system is still needed, there is broad literature about “selfish genetic elements” that can be considered to answer this kind of question. There is a section about selfish genetic elements and hybridogenesis in Quilodran et al. (2018) Conservation Biology.
Thank you so much for the information about selfish genetic elements. We have been discussing about it and now the analysis is on going. So, we added description about the transposable elements in discussion section as described above.
Minor comments:
- Line 21. Could you be more specific about the “essential mechanisms” that remain unclear?
We changed to “the genetic mechanisms” in L22.
- Line 21. You may consider saying: "Here, we focused on four Russian populations"
We changed in L22-23 to “Here, we investigated the water frogs in the western Russian territory.”
- Line 42. Add a reference at the end of the first sentence.
We changed it in L44 as follows:
“Hybridogenesis in an interspecific hybrid mainly occurs in fish and frogs [3−8].”
- Line 55. "Decide" is probably not the best word to used here.
We changed it in L57-58 as follows:
“What genetic factor is involved in choosing the genome to be eliminated?”
- Lines 59-60. Could you delete or modify the sentence: "Therefore, we anticipate new findings in these obscure populations"? In the previous sentence, you have already said that these populations have been poorly studied, which is already sufficient justification for your study.
We deleted the sentence.
- Figure 1. Add a geographical coordinate reference for the top figure.
We added coordinate and also names of rivers in Figure 1.
- Line 112. Please use italics for all scientific names. Check all along the manuscript (e.g., line 116).
Thank you for this. They are our carelessness. We revised all them to italics.
- Figure 3. The font size of the number “0.020” at the bottom left of the figure is too small.
We revised the font size larger. Also, made the bar thicker.
- Line 214 and Line 248. Specify that you are considering "four" Russian populations. However, define earlier in the manuscript if these locations can actually be considered different populations.
As responded to the same earlier comment, we added sentences to explain the definition of population and the reason we unified the locations 3a and 3b to P3 population in figure 1 legend and Materials and methods in L79-82.
- Line 293. "The following figures and tables"?
Yes, it is. We added the words in L324.
- You may consider asking a native English speaker to read the full manuscript.
We asked it. The certification letter is attached.

Reviewer 3 Report
This study is a description of four populations of water frogs from the eastern part of their distribution and for this reason is interesting and fills the gap in our knowledge about these unusual amphibians. However, it is not enough to build a model of "transition". Therefore also the title should be changed. Generally, the conclusions are too-far reaching and basing on scarce dataset.
I have provided my comments directly on the text (attached).

Author Response
#The line numbers in comments are those of the original text, while the lines in our responses are those of the revised text.
Reviewer 3
Our responses to the comments are written in red below.
Comments and Suggestions for Authors
This study is a description of four populations of water frogs from the eastern part of their distribution and for this reason is interesting and fills the gap in our knowledge about these unusual amphibians. However, it is not enough to build a model of "transition". Therefore also the title should be changed. Generally, the conclusions are too-far reaching and basing on scarce dataset.
I have provided my comments directly on the text (attached).
Submission Date
27 November 2020
Date of this review
19 Dec 2020 19:37:19
Responses to comments
Title and abstract
Title
Transition - Variety?
We changed it to: Intrapopulation variation in genome elimination
Abstract
gametogenic cell line
We revised to “gametogenic cell line” in L19.
too far-reaching conclusion
We changed it in L28-29 to:
“We detected a variation in genome elimination within a population.”
Introduction
Mayr
We revised.
To date, few genetic studies have included the Russian populations.
Citation needed
We revised the sentence with added references in L61-63:
“To date, few genetic studies have included the populations (Dedukh et al, 2019) and unusual situation of hybridogenesis in the water frog complex is expected to be brought about by immigration of another levant frog into the Russian territories (Svinin et al., 2015; Ivanov et al., 2015; Zamaletdinov et al., 2015).”
L61-62 too far-reaching conclusion
We revised it to:
L63-64 “Our study identified a variation in the genome elimination within one population.”
Materials and Methods
- Please, specify season (e.g. spring or breeding period),
- only adults or juveniles & adults?
All frogs we investigated are matured adults.
Please see the Table S1 for the details
We changed the sentence in L78-79 to:
“108 adult frogs in total were collected from five locations in the middle of East European plane of Russia in May to September during 2017 to 2019 (Figure 1).”
Citation needed
In addition, identification of P. esculentus was confirmed based on the heterozygous sequences of Sox3 and Rhodopsin genes.
We added Figure S1 in L87 for the reference and example.
Takara, Please, unify
We unified all to TaKaRa.
Citation needed in The primers used were: Cyt b-f 5’ctcctrggagtctgcc3’ and Cyt b-r 5’agrtctttgtaggara3’ for cytochrome b, and SAi1-f5’tgtactggcgacccta3’, and SAi1-r 5’ctgcctttacaatatc3’ for Serum albumin intron 1.
These primers were newly designed by us. So, no citations here.
according to whom? Citation needed.
“according to the manufacturer's instruction.”
It is Applied Biosystems, which is described in the sentence.
what about information from Genebank?
We added these information to the sentence in L113-117:
“Accession Nos. LC599369~LC599378 for cytochrome b and LC599379~LC599385 for Serum albumin intron 1. The sequence data of cytochrome b (MG214959.1, KY613559.1, KU158348.1 and AJ880677.1) and Serum albunin intron 1 (FN432363.1, FN432364.1, FN432365.1, FN432372.1, LN794317.1 and FN432377.1) were obtained from Genbank.”
Ethidium bromide
Revised to ethidium bromide
Species constitution
Revised to “taxonomic composition”
Species names
They are all changed to italics. Sorry for this. These are our carelessness.
Were all the individuals tested? According to Table S1 it seems that not.
Species identification was done in all frogs listed in table S1. The determination of nucleotide sequences of cytochrome b and serum albumin intron 1 was done in some frogs which are marked in table S1 with “Cytb” or “SA”.
Figure 2
Ridibundus were adults or juveniles?
They are adults.
not clearly indicated in Fig. 3:
The haplotype of P. lessonae from Sweden
the haplotype of P. ridbundus from Poland
We made the colored boxes in figure 3 smaller and thus the sequence names from GenBank can be seen more easily because the long name with accession number sticking out from the box.
In figure 3
tu jest błąd, wpisałem ten numer do genbank i ta sekwencja pochodzi z Iranu a nie Polski
Thanks. This is our mistake and then revised to “Iran”.
Table 1
[Fertility] - How was it measured/calculated?
It is a rate of normal tail-bud embryos to the total number of eggs laid.
We added the explanation in table S2.
Triploid: Is it a suspiction or you have another proof?
It is based on the heterozygosity of SAI-1 for the R genome of the esculentus frog. We have no ideas to explain the heterozygosity other than triploidy. In addition, the relative difference in band depth between R and L bands of SAI-1 in the frog is different from those of the other RL: L band is much fainter than R band (please see figure 5-b). The frog was transported to our lab after death. So, we could not check the karyotypes. All others were diploid by karyotyping. Because we have no other data to confirm its triploidy, “may” was changed to “might” as follows in L159:
“One kl. esculentus female (P1-3) was a heterozygous BR, which might be a triploid with the mitochondrial Cyt b of the B haplotype (Table 1).”
Gonadl
It was revised to “gonadal”.
Figure 5 legend
one what?
It was revised in L181 to “On the other hand, one female and one male P. kl. esculentus…”
“Hybridogenesis in testis” should be changed into "spermatogenic cells"
It is revised to “spermatogenic cells” in L202.
As expected by the lower fertility in some P. esculentus frogs, a limited volume
of sperm were observed in the restricted seminiferous tubules.”
please, specify a season (month). Number of spermatozoa depends on time of capture.
We added words as follows in L203-204:
“To investigate spermatogenesis in the testes of P. kl. esculentus, we histologically observed the inner structure of the testes in June (a breeding season).”
Discussions
in the gonads of P. esculentus.
It was revised in L214 to “from gametogenic cells of P. kl. esculentus.
oogoina
It was revised in L225 to “oogonia”
Chmielewska et al.; Dedukh et al.
We added the two references in L227.
Chmielewska M, Dedukh D, Haczkiewicz K, Rozenblut-Kościsty B, Kaźmierczak M, Kolenda K, Serwa E, Pietras-Lebioda A, Krasikova A, Ogielska M. The programmed DNA elimination and formation of micronuclei in germ line cells of the natural hybridogenetic water frog Pelophylax esculentus. Sci Rep. 2018 May 18;8(1):7870. doi: 10.1038/s41598-018-26168-z
Dedukh, D., Litvinchuk, S., Rosanov, J., Shabanov, D., Krasikova, A. Mutual maintenance of di- and triploid Pelophylax esculentus hybrids in R-E systems: results from artificial crossings experiments. BMC Evol. Biol. 2017, 17.
P. esculentus testes eliminates…
It was changed in L235 to “P. esculentus spermatogenic cells eliminates …
always? How do you know?
We removed the word “always” from the sentence in L236.
“Therefore, this population may have extended its distribution east to Russia from the original European populations.”
Why do you think so? The populations might have been there for thousands of years.
As mentioned in the discussion,
it is because we identified the same haplotype of Serum albumin intron1 in P. esculentus from the P1 and P2 as that found in German and Poland populations. We speculate that this haplotype has been succeeded from the original population through clonal inheritance of R genome in P. esculentus. If P. esculentus evolved independently in Russia, this haplotype must not be found, but the genome of P. esculentus should be composed of L and R genomes living there or close to the distributions.
Figure 6 legend
“lessona” mt genome was revised to “lessonae” mt genome.
Thank you so much. This is our big mistake and was revised.
How do you know that it was an introduction?
P. bedriagae lives in Anatolia. Its presence in the middle of Russia needs deeper discussion. It might has been introduced, however, the status of this species is still not clear (see comments: https://amphibiaweb.org/species/5078)
In the papers published by Russian researchers [19-21], they identified mt and nuclear haplotypes of P. bedriage in the populations in the western territories of Russia. So, although we have no ideas about the movement, which was introduced by humans or natural emigration, it is evident that we identified the genetic signature of P. bedriage in P. esculentus and P. ridibundus in the populations we investigated. We do not discuss about how did P. bedriagae move from the original distribution, but do just introgression from P. bedriagae, its genome.
So, we revised the short title to “4.3. Introgression from marsh and levant frogs”.
Are you sure it was triploid?
Please see the above response about triploid.
Too far-going conclusion.
We change the sentences as follows in L266-274:
“We suggest the following hybridogenetic mechanisms existing in the frogs of this western Russian territory: 1) L genome elimination basically does not depend on mitochondrial origins (L mitochondria in P4, while R or B mitochondria in P1 and P2 (except in one male), and 2) the introduction of the marsh frog P. ridibundus and levant frog P. bedriagae into the original L-E system has contributed to the evolution of the L-R-E system (P2). This has produced both sexes of P. kl. esculentus by the introduction of X chromosomes of P. ridibundus or P. bedriagae, allowing RR survival, and 3) the introgression from P. bedriagae changed the hybridogenetic system in one kl. esculentus male, from L genome elimination to L or R genome elimination under lessonae mitochondria (Figure 6). The third observation demonstrates a variation in the genome elimination within a population.”
Triploid
See the above response to the comment
again: why introduction?
Hybridogenetic gametogenesis, not system.
We changed the words to “3) the introgression from P. bedriagae changed the hybridogenetic gametogenesis in one kl. esculentus male, from L genome elimination to L or R genome elimination under lessonae mitochondria (Figure 6).” in L274-276.
How could it happen? It is not possible during a male's live.
What do you mean by "system"?
“3) the introgression from P. bedriagae changed the hybridogenetic system in one kl. esculentus male, from L genome elimination to L or R genome elimination under lessonae mitochondria (Figure 6)..”
Not during a single male’s live, but it took a long time through lives of esculentus males and females since introgression of R or B genome.
Ridibundus and bedrigagae were emigrated into the L-E system populations, and thereby esculentus males and females were born by crossing of RR female x RL male.
What does “system” mean?
We removed the word, and the sentence in L276-277 was revised to “ variation in genome elimination within a population”.
Were they adults or juveniles?
Adults. Please see Table S1.
Again: why do you think that water frogs have dipearsed from Europe to Russia?
Please see the above response to the same comment.
Ragghianti
Sorry for this. We revised.
who dies?
We revised as follows in L295:
and “the hybrids” finally die,
References
klonale, Günther, Rybacki, Heppich-Tunner
Thank you so much for these notes. We revised all.
Round 2
Reviewer 1 Report
Overall, the manuscript significantly improved after the revision.The title and the discussion was tuned down a bit by the authors what might have reduced the soundness but more precisely define the meaning of their findings. Finally, I have only a few minor suggestions listed below.
Line 20: "the genome of one parental species with endoduplication of the remaining parental species" - Please correct this statement. Do you mean the remaining genome of parental species?
Line 74: In the legend of Figure 1 what is the letter 'E' stand for at "E(RL)" ?
Line 102: Correct (Cyp b) to (Cyt b).
Line 125: Correct P. ridbundus to P. ridibundus.
Line 130: The same in Line 125.
Line 132: I suggest that for the first sentence of the legend: "Taxonomic compositon of P. esculentus complex in the sampled populations" instead of "Frog taxonomic composition of the four populations".
Lines 228-229: It is unnecessary to repeat the aim in the discussion
Line 262: Change "Another triploid individual (P1-3) in P1" to "Another possibly triploid individual (P1-3) in P1"
Line 264: Change " P. kl. esculentus triploid male (P1-1)" to " P. kl. esculentus possibly triploid male (P1-1)
Author Response
Dear Reviewer 1,
Thank you for the comments. We responded all the comments and revised the text, which are written in blue.
Overall, the manuscript significantly improved after the revision.The title and the discussion was tuned down a bit by the authors what might have reduced the soundness but more precisely define the meaning of their findings. Finally, I have only a few minor suggestions listed below.
・Line 20: "the genome of one parental species with endoduplication of the remaining parental species" - Please correct this statement. Do you mean the remaining genome of parental species?
It was revised to in L20:
“with endoduplication of the remaining genome of the other parental species.”
・Line 74: In the legend of Figure 1 what is the letter 'E' stand for at "E(RL)" ?
In L73, it was revised to RL, ….
・Line 102: Correct (Cyp b) to (Cyt b).
Thanks. It was revised in L101.
・Line 125: Correct P. ridbundus to P. ridibundus.
Thanks. It was revised in L124.
・Line 130: The same in Line 125.
Thanks. It was revised in L129.
・Line 132: I suggest that for the first sentence of the legend: "Taxonomic compositon of P. esculentus complex in the sampled populations" instead of "Frog taxonomic composition of the four populations".
Thanks. It was changed as suggested in L131.
・Lines 228-229: It is unnecessary to repeat the aim in the discussion
The sentence was removed in L227.
・Line 262: Change "Another triploid individual (P1-3) in P1" to "Another possibly triploid individual (P1-3) in P1"
It was revised to “most probable triploid” in according to the comment of another reviewer in L260.
・Line 264: Change " P. kl. esculentus triploid male (P1-1)" to " P. kl. esculentus possibly triploid male (P1-1)
Revised to “most probable triploid” in L262-263.
Reviewer 3 Report
Dear Authors,
I found your study really interesting, as before. I also accept your corrections. However, I insist to remove the paragraphs, which are purely speculative (marked yellow in the attached file and with my comments). Your research is valuable enough and there is no need to blow it up with speculations that do not fit.
By the way: your speculations are very interesting and you can follow them in your next studies.

Author Response
Dear Reviewer 2,
Thank you for the comments. We responded all the comments and revised the text, which are written in red.
・The best would be "composition". You do not study elimination, you deduce it from the genome compositions.
We want to stay “elimination” because we directly observed that SAI-1 band depth of L genome was reduced, indicating that L genome (gene) was eliminated.  And, production of both R and L sperm from one esculentus male promises elimination of either genome from one germ cell.
・Should be: "We deduced that in 13 RL frogs the L genome should have been eliminated and R was transmitted".
Please, remember that you don't study elimination. You rather deduced transition, which is correct, but should be clearly stated.
We would like to stay “eliminated”.
You are right, but as stated above, we directly observed the reduced depth of L genome in SAI-1. And, it is now almost impossible to prove the genome elimination directly due to difficulty to cytogenetically catch the picture when eliminated.
・remove (males usually have testes)
“testes” was removed in L28.
・speculate, hypothesize
“suggest” was changed to “hypothesis” in L29.
・“Are the two systems stable or changeable with each other? What genetic
factor is involved in choosing the genome to be eliminated?”
“to approach the above questions,”
These are really fundamental and very important questios, which I wish you to answer. But they do not fit to this particular study. This study is valuable enough as a description of new populations and introgressions. Please, remove the sentences.
We hope to stay the sentences in L56-58.
These were revised and changed in the first revised version in according to the comment by one of the 1st reviewers in order to clarify the objective more.
・“To date, few genetic studies have included the populations [18] and unusual situation of hybridogenesis in the water frog complex is expected to be brought about
by immigration of another levant frog into the Russian territories [19−21].”
To be transfered to Discussion. At this point of the text we don't know yet that you have discovered bedriage DNA.
We would like to stay these sentences in L60-62.
Based on the references, bedrigagae genome was actually detected in the western Russian territories. So, we can expect similar things to happen and also something changed in hybridogenesis. In addition, another reviewer in 1streviewing strongly requested us to add these sentences.
・P. bedriage
It was added and “another” was removed: levant frog P. bedriagae … in L62.
・.. variation in genome composition, and thus variation of genome elimination ...
As stated above, we would like to stay this sentence in L63.
・ratio? in table 1.
“rate” was revised to “ratio” in table 1 bottom.
・...whether the remaining genome is transfered to the progeny ...
We revised the words as suggested in L184-185.
・please, remove “in the testes”
It was removed in L202.
・Were
Revised in L208.
・I still don't agree. Why do you think that the frogs have immigrated from western Europe? They might have been in the study site from the very beginning.
This is really too far-reached conclusion.
We described one proof in the next sentence for this question in L238-240 :
“In fact, we identified the haplotype of Serum albumin intron 1 of P. ridibundus that is specific to Poland and Germany in the genomes of P. kl. esculentus (P2-1 and P2-2; Figure 4). “
The haplotype is completely the same as that found in Germany and Poland. This must have been conserved clonally through the esculentus. We do not say all, but a part of descendants coming from Europe. So, we wrote “may include descendants that have extended” in the text in L237-238.
・... a presumend hybridogenetic ... in figure 6 legend
We revised to “Diagram showing presumed hybridogenetic gametogenesis…” in L242 in figure 6 legend.
・most probable triploid
Thanks. It was revised in L260 and 262-263.
・speculate
Changed to “speculate” in L266.
・I still think these are speculations, as I wrote previously:
These are conclusions that go too far and are without strong evidence. You have studied only about one hundred of frogs from 4 populations over two seasons. It is not enough to build a model of transition
We showed one case in the western Russia as you said. We expect next investigations will clarify whether it is widespread or not there. Anyway, this is one report showing variation in genome elimination: we actually found one male and probably his sons too who showed R or L elimination in the population, of which all the other esculentus eliminated L genome only in germ cells.
・provide
revised to “provided” in L297.
・agree, but this conclusion has nothing to do with this particular study. It should be removed. Transposones are very promissing targets of studies on hybridogenesis. I wish you many successes in this field.
We hope to stay the paragraph in L298-307.
This final paragraph was added in 1st revision in order to speculate the mechanisms of hybridogenesis and its variation caused by introgression from P. bedriagae in this study. It was also needed to refer to the issues provided in the introduction such as genome incompatibility and reproductive isolation. One of the first reviewers strongly requested us this logical story running through the text from introduction to discussion.
・to be removed
“in the gonads” was removed in L314.